# Accurate Active and Reactive Power Sharing Based on a Modified Droop Control Method for Islanded Microgrids

**DOI:** 10.3390/s23146269

**Published:** 2023-07-10

**Authors:** Zhi Zhang, Sheng Gao, Caomao Zhong, Zhaoyun Zhang

**Affiliations:** 1Department of Electrical Engineering and Automation, Dongguan University of Technology, Dongguan 523808, China; ztianya2023@163.com (S.G.); zhangzy@dgut.edu.cn (Z.Z.); 2School of Automation, Guangdong University of Technology, Guangzhou 510006, China; 18162171778@163.com

**Keywords:** modified droop control, power sharing, distributed generation unit, islanded microgrid, signal injection

## Abstract

When multiple paralleled distributed generation (DG) units operate in an islanded microgrid, accurate power sharing of each DG unit cannot be achieved with a conventional droop control strategy due to mismatched feeder impedance. In this paper, a small AC signal (SACS)-injection-based modified droop control method is presented for accurate active and reactive power sharing among DG units. The proposed control method adjusts the voltage amplitude of each DG unit by injecting small AC signals to form a reactive power control loop. This strategy does not need communication links or to specifically obtain the physical parameter of the feeder impedance and only requires the local information. Moreover, the parameter design procedure and stability analysis are given full consideration. Finally, simulation and experimental results verify the effectiveness of the proposed control scheme, and accurate active and reactive power sharing can be achieved.

## 1. Introduction

The microgrid, which consists of a variety of distributed generation (DG) units, such as photovoltaic systems, wind power systems, full cells and other energy storage systems, will become an effective supplement to the main power grid and the potential energy structure [1,2,3]. These renewable DG units have the advantages of reductions in the pollution caused by fossil fuels, decreased power transmission losses, ease of installation, and so on. However, the high level of penetration of renewable DG units will introduce serious challenges to the main power grid or microgrid, such as frequency and voltage deviations and power sharing and fluctuation [4,5]. In an islanded microgrid in particular, multiple voltage source converter (VSC)-based DG units paralleled together should not only provide voltage and frequency support for the loads but should also achieve accurate power sharing according to their power rating [6,7,8].

The voltage and frequency droop control method is widely adopted for the parallel connection of multiple inverter-based DG units, and accurate active power sharing can be achieved in a steady state because the feeder impedance is usually mostly inductive [1]. However, due to the coupling of active and reactive power and mismatched feeder impedance, high performance cannot be guaranteed by the conventional droop control method with respect to reactive power sharing [9,10,11]. Various improved droop control strategies have been proposed, and they are mainly divided into two categories: communication-based improved droop control techniques [12,13,14,15,16,17,18] and communication-less improved droop control techniques [19,20,21,22,23,24,25,26,27].

High voltage and frequency regulation performance and acceptable power sharing can be obtained via communication-based improved droop control techniques in islanded microgrids, and these improved methods can be divided into several families: hierarchical control [12,13,14], distributed control [15] and successive-approximation-based control [16,17,18]. The hierarchical control method of the microgrid was proposed in [12,13], and a microgrid central controller (MGCC) was used to obtain the reactive power of each DG unit, calculate the compensator voltage and send back the control information to each DG unit via a low-bandwidth communication system; however, in this scenario, the MGCC and the communication line must be highly reliable to ensure that the microgrid works normally. The distributed control method, which integrates the primary and secondary controllers together as a local controller, was proposed to enhance the reliability of the microgrid [15], and in this method, the failure of one DG would not affect the normal operation of other DG units; however the drawbacks of communication delays and the cost of the communication lines remain [16,17,18]. A virtual impedance tuning method based on successive approximation was proposed to compensate for the mismatched feeder impedance [16], and in this method, only local information is required to achieve accurate reactive power sharing; however, a common triggering signal that is sent from the microgrid center to each DG unit is still needed to establish the internal time sequence. Similarly, a two-stage adaptive virtual resistor control scheme combining a synchronous maximum power bus (SMPB) was presented in [18] to eliminate fundamental power and harmonic power-sharing errors. A similar approach was also proposed in [17] to achieve accurate active power and harmonic power sharing, but reactive power sharing was not considered. Although the above-mentioned methods proposed in [16,18] for each DG unit do not require real-time information transmitted by the communication line, the power-sharing performance is still significantly dependent on a common triggering signal which is still connected between DG units.

The improved droop control method without communication links has the advantages of a low cost, easy extensibility, a plug and play feature and high reliability. In order to solve the problem of mismatched feeder impedance, various control strategies based on virtual impedance have been proposed for fundamental power sharing [19,20,21,22,23,24,25,26,27]. The virtual inductor control method was proposed in [19] to enhance the reactive power sharing of parallel DG inverters; however, the sharing error still cannot be eliminated. The authors of [20] also used the virtual inductor for accurate power control in a low-voltage microgrid, but the feeder impedance should be first estimated in grid-connected mode. An adaptive virtual impedance was proposed in [21] to improve the reactive power-sharing accuracy using only local information, and the sharing error could be reduced by approximately 50% compared to conventional droop control. An enhanced virtual impedance control scheme was proposed in [22] for accurate power sharing, but the voltage of the point of common coupling (PCC) must be measured, which is impractical in real situations. A virtual capacitor-based control algorithm was proposed in [23] for reducing the reactive power sharing error; this algorithm simulates the characteristics of the paralleling capacitor at the DG unit output terminal, and an adaptive coefficient *n_c_* is used to adjust the value of the virtual capacitor. However, with the increase in the coefficient, although the error of reactive power sharing decreases, the stability of the system also decreases [24]. The authors of [24,25,26,27] proposed an improved control strategy combining virtual impedance and virtual capacitance to achieve fast reactive power sharing and a low circulating current between inverters and to improve the reactive power sharing accuracy and system stability. However, accurate reactive power sharing still cannot be achieved. An adaptive regulation droop coefficient control method was proposed in [25] for photovoltaic microgrid systems to improve active power sharing. A virtual parallel inductor (VPI) concept was also proposed in [26] to minimize the reactive power sharing error in a virtual synchronous generator-controlled microgrid.

All the aforementioned improved droop control methods without communication links have the following characteristics: (1) most of the control strategies can only reduce the reactive power sharing error; and (2) for accurate reactive power sharing, additional information must be obtained or measured, which may be impractical in most cases.

In addition, some other methods without communication lines have been proposed [28,29]. A droop control strategy was proposed in [28] to improve reactive power sharing in which an integral term is used in the *Q*-*V* droop controller to restore the voltage amplitude. However, the sharing accuracy cannot be guaranteed during the restoration process [16]. A distributed optimal control strategy based on a Kalman filter state estimator was proposed for performing reactive power sharing and system voltage restoration via local measurement [29], but it requires a solution for the optimal regulators to be achieved by computing an optimization cost function. The high computational burden is not suitable for industrial controllers, and it is not feasible for practical microgrid application scenarios [4].

In [30], a small AC signal (SACS)-injection-based control method for achieving frequency restoration and accurate active power sharing was proposed; however, reactive power sharing was not considered. Therefore, a SACS-injection-based modified droop control method is proposed in this paper for accurate active and reactive power sharing. The main contributions of the proposed control strategy can be summarized as follows: (1) since no communications lines are needed, a low-cost and highly adaptable microgrid configuration can be achieved; (2) the specific parameter of the feeder impedance does not need to be known; and (3) accurate fundamental active and reactive power sharing can be achieved with only local information. Table 1 shows a comparison of various control methods.

This paper is organized as follows. In Section 2, a brief introduction of the conventional droop control method is provided, and a SACS-injection-based modified droop control strategy for accurate power sharing is proposed in Section 3. In Section 4, the parameter design procedure and stability analysis are described in detail. Simulation and experimental results are provided in Section 5 to verify the correctness and effectiveness of the proposed droop control method. Finally, the conclusions of this paper are presented in Section 6.

## 2. Review of the Conventional Droop Control Method

Figure 1 shows the simplified structure of the islanded microgrid with multiple DG units in parallel operation, and the block diagram with the conventional droop control method for power sharing is illustrated in Figure 2.

### 2.1. Outer Droop Control Loop

In islanded mode, the inverter-based DG units work as voltage source converters (VSCs) in parallel to provide voltage and frequency support for the microgrid, and all DG units with different power capacities must achieve power sharing according to their own rated power for the economical and reliable operation of the microgrid. As shown in Figure 2, each DG unit calculates its own active and reactive power by measuring the local output voltage *U*_c_ and current *I*_o_. The conventional *P*-*ω* and *Q*-*E* control algorithm, as described in Equations (1) and (2), is also implemented for load power sharing and as the synthetic voltage reference. The mathematical expression of the droop control method can be shown as:(1)$$\omega_{ref}=\omega_{0}-k_{p}(P-P_{0})$$
(2)$$E_{ref}=E_{0}-k_{q}(Q-Q_{0})$$
where *P* and *Q* are the output active and reactive power of the inverter, respectively. *P*_0_ and *Q*_0_ are the reference active and reactive power, respectively, and they are usually set to zero in islanded microgrids. *ω_ref_* and *E_ref_* are the synthetic reference frequency and voltage amplitude, respectively. *ω*_0_ and *E*_0_ are the nominal frequency and voltage amplitude, respectively. *k_p_* and *k_q_* are the droop coefficients for the inverter-based DGs operating in islanded mode, and the values are chosen according to the allowable maximum frequency and voltage amplitude deviation of the inverter. When multiple DGs with different power ratings are connected in parallel, the droop coefficients should be designed to share the load power in proportion to each rated power [19].
(3)$$\left\{\begin{array}{c} k_{p1}P_{1}=k_{p2}P_{2}=\cdots =k_{pN}P_{N}\\ k_{q1}Q_{1}=k_{q2}Q_{2}=\cdots =k_{qN}Q_{N} \end{array}\right.$$
where *k_pn_* and *k_qn_* (*n* = 1, 2, ..., *N*) are the droop coefficients corresponding to the *n*th DG unit, and *P_n_* and *Q_n_* are the nominal active and reactive power of each DG, respectively.

Notably, Equation (1) holds only if the line impedance between the PCC and the DG unit is mainly inductive, as shown in Figure 3, and the active and reactive power flow from the DG unit to the PCC can be expressed as follows [31]:(4)$$\left\{\begin{array}{c} P=\frac{1}{R^{2}+X^{2}}(RE^{2}-REU_{p}\cos\delta +XEU_{p}\sin\delta )\\ Q=\frac{1}{R^{2}+X^{2}}(XE^{2}-XEU_{p}\cos\delta -REU_{p}\sin\delta ) \end{array}\right.$$
where *E* and *U_p_* are the output voltages of the DG and PCC voltage amplitudes, respectively; *X* and *R* represent the feeder reactance and resistance, respectively; and *δ* is the power angle difference between the DG and PCC voltages. When *X* >> *R*, Equation (4) can be rewritten as:(5)$$\left\{\begin{array}{l} P\approx \frac{EU_{p}\sin\delta }{X}\\ Q\approx \frac{E(E-U_{p}\cos\delta )}{X} \end{array}\right.$$

Usually, the power angle *δ* is small, so it satisfies *sinδ* ≈ *δ* and *cosδ* ≈ 1, and Equation (5) can be simplified as:(6)$$P\approx \frac{EU_{p}\delta }{X}$$
(7)$$Q\approx \frac{E(E-U_{p})}{X}$$

Moreover, when the reactance is much less than the resistance of the line impedance *X* << *R*, the delivered active and reactive power can be approximated as:(8)$$P\approx \frac{E(E-U_{p}\cos\delta )}{R}\approx \frac{E(E-U_{p})}{R}$$
(9)$$Q\approx \frac{-EU_{p}\sin\delta }{R}\approx \frac{-EU_{p}\delta }{R}$$

### 2.2. Inner Voltage Regulation Loop

The reference voltage is obtained through the above droop controller, and then the output voltage of each DG unit is generated based on the voltage regulation loop. The inner voltage regulation loop also consists of an outer voltage loop and an inner current loop based on the *αβ* stationary frame, as shown in Figure 4. *G_V_*(*s*) and *G_I_*(*s*) are the transfer functions of the voltage-loop controller and the current-loop controller, respectively, and they can be expressed as [14]:(10)$$\left\{\begin{array}{l} G_{V}(s)=k_{pv}+\sum_{h=1,5,7,9}\frac{2k_{ivh}s}{s^{2}+2\omega_{cvh}s+\omega_{h}^{2}}\\ G_{I}(s)=k_{pi} \end{array}\right.$$
where *G_V_*(*s*) adopts a proportional resonant (PR) controller, *G_I_*(*s*) uses a proportional controller, *k_pv_* and *k_pi_* are the proportional gains, *k_ivh_* is the fundamental or harmonic resonant gain term, *ω_cvh_* is the cutoff frequency of the resonant controllers and *ω_h_* is the resonant frequency.

When the feeder impedance is mainly an inductor, the conventional droop technique can achieve accurate active power sharing; however, it has poor reactive power sharing performance due to the mismatch of the feeder impedance [1,4]. In the following section, a modified droop control strategy based on SACS injection is proposed for accurate active and reactive power sharing without communication links.

## 3. Proposed SACS-Injection-Based Modified Droop Method for Power Sharing

In this section, a SACS-injection-based modified droop control method without communication links is proposed which is motivated by the aforementioned secondary control strategy [14,30,32], and a detailed introduction is illustrated in the following section.

### 3.1. Motivated by the Secondary Control Strategy through PI Controller for Power Sharing

The authors of [14,32] proposed a secondary control strategy for accurate reactive power sharing. First, the secondary controller obtains the output reactive power *Q* of each DG unit via communication links, and then the reference reactive power *Q** is calculated and sent back to the primary control. Finally, accurate reactive power sharing can be achieved by adding the compensator Δ*E*, calculated through the PI controller to the nominal voltage amplitude *E*_0_, which is provided in the following equations:(11)$$\left\{\begin{array}{l} \omega_{ref}=\omega_{0}-k_{p}(P-P_{0})\\ E_{ref}=E_{0}-k_{q}(Q-Q_{0})+\Delta E \end{array}\right.$$
where Δ*E* = *k_ps_* (*Q** − *Q*) + *k_is_*/s (*Q** − *Q*), and *k_ps_* and *k_is_* are the proportion and integration gains, respectively. Equation (11) can also be rewritten as
(12)$$E_{ref}=E-k_{q}(Q-Q_{0})$$
where *E* = *E*_0_ + Δ*E*.

As shown in Figure 5, the output reactive power *Q* of each DG unit can be adjusted due to the voltage compensator Δ*E*, and it can also achieve the reference value *Q** due to the existence of the integral term *k_is_*/s. However, communication line failure and communication delay may result in poor reactive power sharing.

### 3.2. SACS-Injection-Based Modified Droop Control Method for Power Sharing

The above-mentioned method can achieve active reactive power sharing by adding the voltage compensator Δ*E* to the *Q*-*E* droop expression; however, communication lines are needed, which increase the complexity and high cost of the system. In this section, a SACS-injection-based modified droop control method, which is similar to Equation (11), is proposed. The difference between the method and equation is that the compensator Δ*E* is composed of the reactive power of the injected SACS, and it can be expressed as follows:(13)$$\begin{array}{ll} E_{ref} & =E_{0}-k_{q}(Q-Q_{0})+\Delta E\\ & =E_{0}-k_{q}(Q-Q_{0})+G_{q}Q_{ss}\\ & =E_{0}^{\prime }-k_{q}(Q-Q_{0}) \end{array}$$
where *E*′_0_ = *E*_0_ + *G_q_
Q_ss_*, and *Q_ss_* is the reactive power of the injected SACS, which will be described in detail in the following section. *G_q_* is the amplifier gain, which is used for amplifying the reactive power of the SACS.

To achieve accurate reactive power sharing for each DG unit, the following droop control strategy for the injected SACS is also used:(14)$$\omega_{ss}=\omega_{ss0}+k_{sq}(Q-Q_{0})$$
where *ω_ss_* and *ω_ss_*_0_ are the reference frequency and the nominal frequency of the injected SACS, respectively, and *k_sq_* is the SACS droop coefficient for reactive power sharing.

If two DG units are connected in parallel, according to droop Equation (14), the frequency difference Δ*ω*_ss_ in the injected SACSs for the two DG units can be expressed as follows:(15)$$\Delta \omega_{ss}=\omega_{ss1}-\omega_{ss2}=k_{sq}(Q_{1}-Q_{2})$$
where *ω_ss_*_1_ and *ω_ss_*_2_ represent the injected SACS frequency of each DG unit. By integrating the frequency difference Δ*ω_ss_*, the phase difference *δ_ss_* of the injected SACSs of each DG can be deduced as
(16)$$\delta_{ss}=\int \Delta \omega_{ss}dt=k_{sq}\int (Q_{1}-Q_{2})dt$$

### 3.3. Overall Control Block Diagram

The overall block diagram of the proposed modified droop control strategy based on the SACS injection method for accurate fundamental active and reactive power sharing is illustrated in Figure 6.

First, only the local signals of the output current *i_oαβ_* and filter capacitor voltage *u_cαβ_* for each DG unit are measured, and then the fundamental current *i_oαβf_* and SACS current *i_oαβss_* are separated via signal extraction [14,30]. The fundamental active power *P*, reactive power *Q* and SACS reactive power *Q_ss_* can be calculated based on the *αβ* frame and are as follows:(17)$$\left\{\begin{array}{l} P=\frac{3}{2}\frac{\omega_{cp}}{s+\omega_{cp}}(u_{c\alpha f}i_{o\alpha f}+u_{c\beta f}i_{o\beta f})\\ Q=\frac{3}{2}\frac{\omega_{cp}}{s+\omega_{cp}}(u_{c\beta f}i_{o\alpha f}-u_{c\alpha f}i_{o\beta f})\\ Q_{ss}=\frac{3}{2}\frac{\omega_{cp}}{s+\omega_{cp}}(u_{c\beta ss}i_{o\alpha ss}-u_{c\alpha ss}i_{o\beta ss}) \end{array}\right.$$
where *u_cαf_*, *u_cβf_*, *i_oαf_* and *i_oβf_* are the output fundamental voltage and current components based on the αβ frame, respectively. *u_αss_*, *u_βss_*, *i_oαss_* and *i_oβss_* are the voltage and current components of the injected SACSs based on the *αβ* frame.

Compared with the conventional droop control method, the proposed modified droop control strategy adds a compensator Δ*E* to the voltage droop control loop, and an extra reference voltage *u*_αβ_ss_* generated by the injected SACS is added to the fundamental reference voltage *u*_αβ_f_* to synthesize the overall reference voltage *u*_αβ_sum_*, which is shown as follows:(18)$$u_{\alpha \beta_{-}sum}^{*}=u_{\alpha \beta_{-}f}^{*}+u_{\alpha \beta_{-}ss}^{*}$$

Meanwhile, to achieve zero-steady-error tracking of the SACS reference voltage, the proportional resonance controller *G_V_*(*s*) of the inner voltage regulation loop should be modified to:(19)$$\begin{array}{c} G_{V}(s)=k_{pv}+\sum_{h=1,5,7,9}\frac{2k_{ivh}s}{s^{2}+2\omega_{cvh}s+\omega_{h}^{2}}\\ +\frac{2k_{ivs}s}{s^{2}+2\omega_{cvs}s+\omega_{ss}^{2}} \end{array}$$
where *k_ivs_* and *ω_cvs_* represent the gain and cutoff frequency of the SACS resonant controller, respectively. To ensure that Equation (14) holds, a virtual resistor is also applied to the feeder impedance to make the feeder impedance mainly resistant for the implementation of SACS frequency droop control. The application of the virtual resistor in the droop control method has been mentioned in many studies [1,19], and it will not be introduced in detail here. Importantly, the overall control system may have virtual reactance and virtual resistance control loops. If the feeder impedance for the fundamental voltage is not mainly inductive, a virtual reactance control loop must be added to ensure that Equations (1) and (2) hold. In most cases, the feeder reactance for the fundamental frequency is mainly inductive, and the virtual reactance does not need to be added. However, the virtual resistor must be added to make the feeder impedance for the injected SACS frequency mainly resistive.

## 4. Parameter Design and Stability Analysis

In this section, all the parameters, such as the frequency *ω_ss_* and amplitude *E_ss_*_0_ of the injected SACS, the amplified gain *G_q_*, and the droop coefficients *k_p_*, *k_q_* and *k_ssq_,* are discussed in detail.

### 4.1. Design of the SACS Frequency and Amplitude

The selection of the SACS frequency and amplitude are critical for the accurate control of active and reactive power sharing. A SACS with a large amplitude is easy to extract, but it will result in a high harmonic component of the output voltage. In contrast, a small voltage amplitude will make an SCAS difficult to extract. Therefore, the voltage amplitude of the injected SACS should be chosen based on the tradeoff between the signal extraction and voltage quality standards [33]. Finally, the amplitude of the injected SACS is chosen to be nearly 1% of the fundamental voltage amplitude.

The frequency of the injected SACS should also be chosen carefully. On the one hand, the frequency of the injected SACS must be different from the output voltage harmonic frequency to easily extract it. It is known that the output voltage contains only odd-order components, whether linear or nonlinear loads. To facilitate signal extraction, we chose the even component as the frequency of the SACS. On the other hand, the output end of the inverter-based DG is usually connected via an *LC* second-order low-pass filter to alleviate the high-order harmonic voltage components; therefore, the frequency of the injected SACS must be lower than the cutoff frequency of the *LC* filter. Finally, the SACS frequency is set to four times the fundamental voltage frequency [30].

### 4.2. Design of the Gain G_q_

Gain *G_q_* is not only related to the SACS power injected into the microgrid system but also affects the stability of the whole system. On one hand, according to the expression of the voltage compensator Δ*E* = *G_q_ Q_ss_*, a small gain *G_q_* leads to the provision of a large SACS reactive power to the system and causes a large harmonic output current. On the other hand, a large gain *G_q_* with small SACS reactive power will lead to the instability of the whole system. Taking two DG units in parallel as an example, the fundamental voltage amplitude difference of two DG units can be expressed as
(20)$$\begin{array}{ll} \Delta E_{12} & =E_{1}-E_{2}\\ & =-k_{q}(Q_{1}-Q_{2})+G_{q}(Q_{ss1}-Q_{ss2}) \end{array}$$
where *E_i_*, *Q_i_* and *Q_ssi_* are the output fundamental voltage amplitude, reactive power and injected SACS reactive power of each DG unit, respectively, where *i* = 1, 2. In a steady state, *Q*_1_ = *Q*_2_, so the above equation can be simplified to:(21)$$G_{q}=\frac{\Delta E_{12}}{Q_{ss1}-Q_{ss2}}=\frac{\Delta E_{12}}{\Delta Q_{ss}}$$
where Δ*Q_ss_* represents the difference in the reactive power of the injected SACSs.

Similar to the equivalent circuit at the fundamental frequency, the two DG units with SACSs are connected in parallel, as shown in Figure 7, and the following equation holds:(22)$$\left\{\begin{array}{c} Q_{ss1}=\frac{Q_{ssL}}{2}+\frac{Q_{ss1}-Q_{ss2}}{2}\\ Q_{ss2}=\frac{Q_{ssL}}{2}-\frac{Q_{ss1}-Q_{ss2}}{2} \end{array}\right.$$
where *Q_ssL_* is the total load power of the reactive power of the injected SACSs. Moreover, 1/2(*Q_ss_*_1_ − *Q_ss_*_2_) represents the SACSs’ reactive power flow from DG1 to DG2.

According to Equation (14) and the realization for the high performance of reactive power sharing, a virtual resistor is adopted to make the feeder impedance mainly resistive for the SACS, which is shown in Figure 8. *E_ssα_**, *E_ss__β_**, *E_ss__α_*_0_ and *E_ss__β_*_0_ represent the reference voltages and the nominal voltage amplitudes of the injected SACSs, respectively.

*R_v_* represents the value of the virtual resistance. Due to the addition of virtual resistance, the feeder impedance of the two SACS DG units is mainly resistive, and the SACS reactive power flows from DG1 to DG2 can also be expressed as:(23)$$Q_{ss12}=\frac{Q_{ss1}-Q_{ss2}}{2}\approx -\frac{E_{ss1}E_{ss2}}{R_{ss1}+R_{ss2}}\sin\delta_{ss}$$
where *E_ssi_* and *R_ssi_* represent the SACS voltage and line impedance, respectively, where *i* = 1, 2. *δ_ss_* is the phase angle difference of the injected SACSs for the two DG units. Moreover, the difference in the reactive power of the injected SACSs is provided by:(24)$$\Delta Q_{ss}=Q_{ss1}-Q_{ss2}=2Q_{ss12}\approx -\frac{2E_{ss1}E_{ss2}}{R_{ss1}+R_{ss2}}\sin\delta_{ss}$$

By substituting Equation (24) into Equation (21), the following expression can be derived as follows:(25)$$Gq=\frac{\left|\Delta E_{12}\right|}{\left|\Delta Q_{ss}\right|}\approx \frac{\Delta E_{12}R_{ss}}{2E_{ss1}E_{ss2}\sin\delta_{ss}}$$
where *R_ss_* = *R_ss_*_1_ + *R_ss_*_2_. According to the power angle stability criteria [34], the power angle *δ_ss_* should not exceed 90°. To maintain a sufficient stability margin, the power angle *δ_ss_* should be less than 45, and *sinδ_ss_* ≈ *δ_ss_* should be true. The minimum value of the gain *G_q_* can be derived as
(26)$$Gq=\frac{\left|\Delta E_{12}\right|}{\left|\Delta Q_{ss}\right|}\approx \frac{\Delta E_{12}R_{ss}}{2E_{ss1}E_{ss2}\sin\delta_{ss}}$$

To simplify the calculation, the amplitudes *E_ss_*_1_ and *E_ss_*_2_ of the SACSs can be approximated to *E_ss_*_0_, and the voltage amplitude difference Δ*E*_12_ is the allowable deviation of the rated voltage *E*_0_.

### 4.3. Design of the Droop Coefficienst k_p_, k_q_ and k_ssq_

With larger droop coefficients of *k_p_* and *k_q_*, the active and reactive power sharing performance will be better, but the frequency and amplitude of the fundamental reference voltage will deviate greatly from the nominal values. The droop parameters *k_p_* and *k_q_* can be designed according to the following criteria [19]:(27)$$k_{p}=\frac{\Delta \omega }{P_{max}}$$
(28)$$k_{q}=\frac{\Delta E}{Q_{max}}$$
where *P_max_* and *Q_max_* are the maximum output of the active and reactive power of each DG unit, and Δ*ω* and Δ*E* are the maximum allowable deviations in the frequency and voltage amplitude.

The dynamic performance of accurate reactive power sharing depends on the droop parameters *k_ssq_* of the injected SACSs. A large droop parameter *k_ssq_* can accelerate the process of accurate power sharing, but it may cause system instability. In contrast, with a small droop parameter *k_ssq_*, the transient process of reactive power sharing can no longer achieve power sharing. Therefore, if the system is stable, a large droop parameter *k_ssq_* should be selected to improve the dynamic performance of the system, and the effect on the stability of the system will be described in detail in the following section.

### 4.4. Stability Analysis

In this section, the relationship between the parameter *k_sq_* and the stability of multiple parallel DG systems will be analyzed in detail. According to Equations (15), (20) and (24), the fundamental voltage amplitude difference of two DG units can be expressed as
(29)$$\begin{array}{ll} E_{1}-E_{2} & =-k_{q}(Q_{1}-Q_{2})+G_{q}(Q_{ss1}-Q_{ss2})\\ & =-(k_{q}\frac{1}{k_{sq}}\delta_{ss}^{\prime }+\frac{2G_{q}E_{ss1}E_{ss2}}{R_{ss}}\delta_{ss}) \end{array}$$

Since low-pass filters are usually used for measuring blocks, Equation (29) can be rewritten as
(30)$$\Delta E_{12}=-\frac{\omega_{c}}{s+\omega_{c}}(k_{q}\frac{1}{k_{sq}}\delta_{ss}^{\prime }+2G_{q}E_{ss1}E_{ss2}R_{ss}\delta_{ss})$$
where *ω_c_* is the cutoff frequency of the low-pass filter.

It is assumed that the total fundamental reactive power of the load is *Q_L_*, and the injected SACS frequency of each DG are equal in the steady state, which satisfies *ω_ss_*_1_ = *ω_ss_*_2_. According to Equation (14), it can be deduced that the output reactive power of the two DG units is equal in the steady state, and it satisfies that *Q*_1_ = *Q*_2_ = *Q_L_*/2. However, when the system is in a transient process, such as load or output fundamental voltage changes, part of the reactive power flows into the load, and another part of the reactive power flows into other DG units. They can be expressed as follows:(31)$$Q_{1}=\frac{Q_{L}}{2}+\frac{Q_{1}-Q_{2}}{2}$$
(32)$$Q_{2}=\frac{Q_{L}}{2}-\frac{Q_{1}-Q_{2}}{2}$$

Since the feeder impedance for the fundamental voltage is usually mainly inductive, and according to Equation (7), the reactive power flowing from DG1 to DG2 can be deduced as
(33)$$\frac{Q_{1}-Q_{2}}{2}\approx \frac{E_{1}(E_{1}-E_{2})}{X_{1}+X_{2}}\approx \frac{E_{0}\Delta E_{12}}{X_{1}+X_{2}}$$

Since the low-pass filter is also used for the power calculation, the above equation can be rewritten as
(34)$$\Delta Q=Q_{2}-Q_{1}\approx \frac{\omega_{c}}{s+\omega_{c}}-\frac{-2E_{0}\Delta E_{12}}{X_{1}+X_{2}}$$

Based on the previous analysis, the control block diagram of reactive power sharing can be derived as shown in Figure 9, and the simulation parameters of the islanded microgrid are listed in Table 1. The root locus of the open-loop transfer function can be obtained as shown in Figure 10. Different color lines represent different root locus and the direction of the arrows indicate that the value of *k* increases in Figure 10. It can be seen that when the value of droop coefficient *k_sq_* is greater than 0.03, the root of the closed-loop system will appear in the right half plane and become unstable. Figure 11 also shows the Bode diagram of the open-loop transfer function, and with the increase in the droop coefficient *k_sq_*, the phase margin of the system decreases gradually. Therefore, the droop coefficient *k_sq_* should be chosen based on the tradeoff between the system stability and dynamic response, and a large coefficient is selected preferentially under the condition of ensuring the stability of the system.

## 5. Simulation and Experimental Results

In this section, the performance of the proposed SACS-injection-based modified droop control strategy is validated based on simulation and experimental results. The detailed system parameters are listed in Table 2.

### 5.1. Simulation Results

The simulation modes of the DG units with the same three-phase full-bridge topology are built based in MATLAB, and three DG units are connected in parallel with different feeder impedances, as shown in Figure 1. Figure 12 demonstrates the simulation waveforms of the output active power *P*, reactive power *Q* and fundamental frequency f for each DG unit via the conventional droop control strategy. In the initial state, *R*_1_ is connected into the islanded microgrid as the load. At 3 s, the load *R*_2_ is also plugged into the microgrid system, and the feeder impedance is mainly inductance. It can be observed that accurate active power sharing can be achieved with the conventional droop control strategy, but the output reactive power for each DG unit is different from the mismatched feeder impedance.

The simulation waveforms of the output active power *P*, reactive power *Q*, fundamental frequency *f*, voltage compensator Δ*E* and SACS frequency *f_ss_* with the proposed modified droop control strategy for the paralleled three DG units are shown in Figure 13, and it can be seen that high-performance power sharing can be achieved. Since the *P − f* droop expression remains unchanged, accurate active power can also be achieved in a steady state. Meanwhile, with the SACS-injection-based modified droop method, the reactive power error for each DG unit can be eliminated. When the load *R*_2_ is also plugged into the microgrid system at 3 s, the reactive power output of these three DG units can also be divided equally quickly. Moreover, the voltage compensator Δ*E*, which is composed of the SACS reactive power *Q_ss_* and gain *G_q_* for each DG unit, is also shown in Figure 13. With different compensators Δ*E*, the conventional *Q-E* droop expression is modified as shown in Figure 5, and the output voltage amplitude of each DG unit is also adjusted. Accurate reactive power will be achieved when the system reaches the steady state.

### 5.2. Experimental Verification

Figure 14 shows the experimental prototype of two paralleled DG units connected in an islanded microgrid, and a three-phase two-level full-bridge PWM inverter is chosen as the main circuit topology for each DG unit. The control algorithm for the DG unit is implemented in the dSPACE SCALEXIO platform (dSPACE, Paderborn, Germany) as shown in Figure 15. A 14-bit analog-to-digital converter (ADC DS6221) (dSPACE) is used to sample the output voltage, inductor current and load current signals of each DG unit at a 20 kHz sampling frequency. The I/O board card (DS6202) (dSPACE) is configured as a PWM module with a 20 kHz switch frequency. A power analyzer (Tek-PA3000) (Tektronix, Inc. Beaverton, OR, USA) is used to measure the output power and voltage signals for analysis, and an oscilloscope (YOKOGAWA-DLM2024) (YOKOGAWA, Tokyo, Japen) is used to collect the output voltage, frequency and other information.

Figure 16 shows the experimental waveform of the output power with the conventional droop control and the proposed SACS-injection-based droop control strategy. In the initial state, the DG units adopt the conventional droop control strategy, and 10 s later, the proposed modified droop control method is applied to the two DG units. Figure 16A demonstrates that the active power can be shared well with both control methods, and Figure 16B shows that the reactive power sharing error can be eliminated with the proposed modified droop method, which is enabled at 10 s.

The dynamic experimental waveform of power sharing with the SACS injection-based modified droop control strategy is shown in Figure 17. The resistive–inductive load *Z*_1_ is connected to the islanded microgrid, and a resistive load *R*_2_ is switched ON or OFF every 8 s. The high performance of the transient process for power sharing is exhibited, and accurate active and reactive power sharing can be achieved in a steady state.

Moreover, the voltage waveform of the PCC is also measured, as shown in Figure 18, and the total harmonic distortion (THD) value of the voltage waveform is also analyzed. Due to the injection of a four-order SACS, a slight voltage distortion is shown in the output voltage waveform, and the THD value of the PCC voltage is 1.30%. However, it can still meet the voltage harmonic standard [29].

## 6. Conclusions

In this paper, a SACS-injection-based modified droop control method is proposed for accurate active and reactive power sharing when DG units operate in islanded microgrids, and the parameter design procedure and stability analysis are described in detail. The proposed modified droop method has the advantage of no communication lines and the specific parameters of feeder impedance, a low cost, the high adaptability of the microgrid configuration and plug-and-play functionality. Finally, simulation and experimental results verify the correctness and effectiveness of the proposed modified droop control method.

However, the method proposed in this paper has some disadvantages: (1) it is necessary to inject small-AC-voltage signals into the system, which will lead to an increase in the harmonic control of the grid voltage, so they must be carefully selected. (2) This method can achieve accurate active power and reactive power sharing, but it can not achieve secondary frequency recovery, which is a direction for the future research.

## Figures and Tables

**Figure 1 sensors-23-06269-f001:**
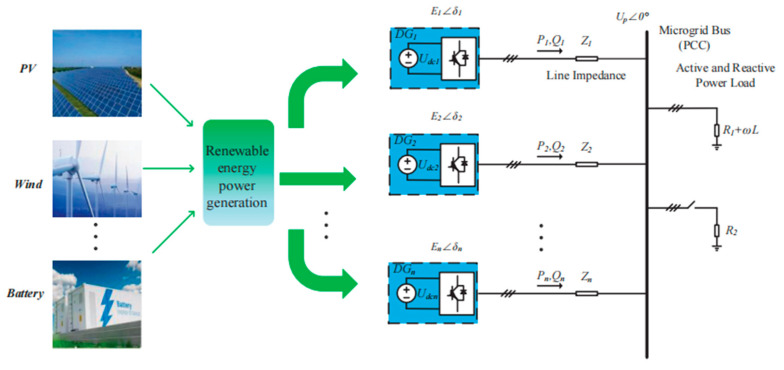
The simplified structure of the islanded microgrid with multiple DG units.

**Figure 2 sensors-23-06269-f002:**
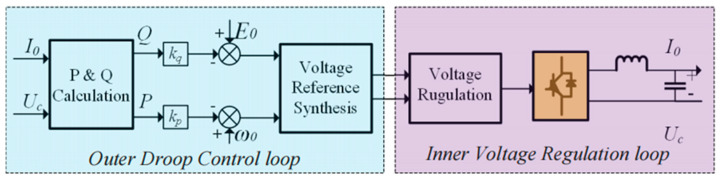
Block diagram with the conventional droop control method.

**Figure 3 sensors-23-06269-f003:**
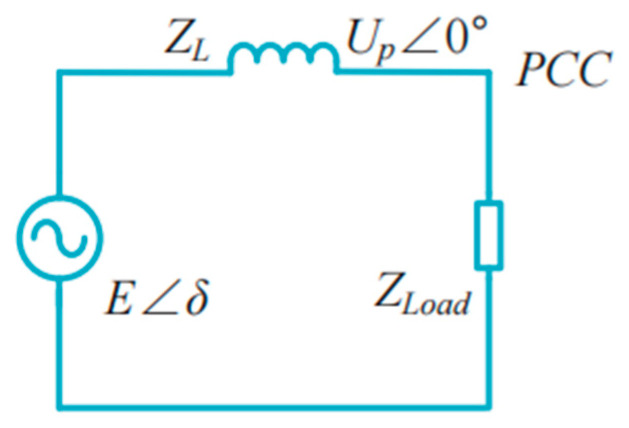
Equivalent circuit of the DG unit connected to the PCC.

**Figure 4 sensors-23-06269-f004:**
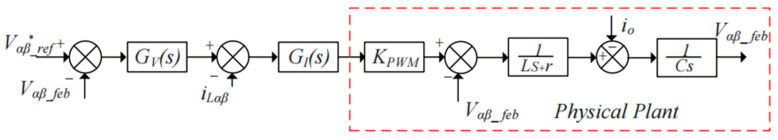
Block diagram of the inner voltage regulation loop.

**Figure 5 sensors-23-06269-f005:**
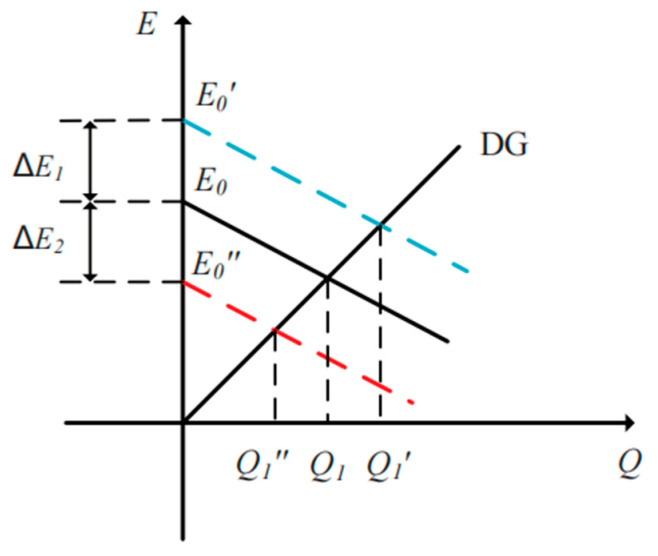
The output reactive power *Q* of each DG unit with different voltage compensators Δ*E*.

**Figure 6 sensors-23-06269-f006:**
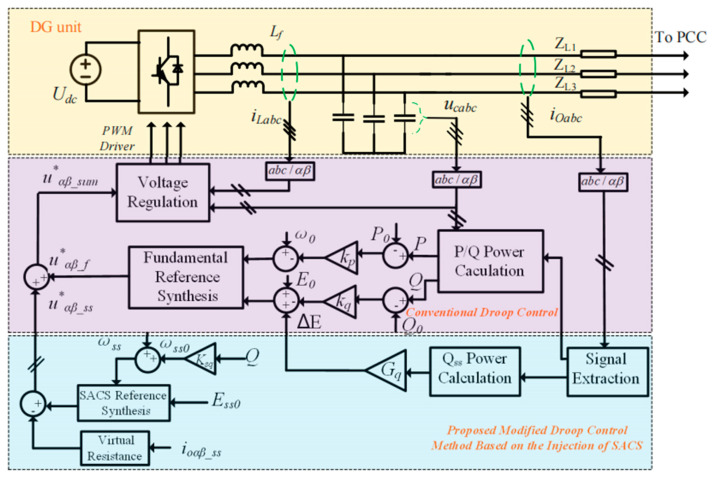
The overall block diagram of the proposed modified droop control strategy.

**Figure 7 sensors-23-06269-f007:**
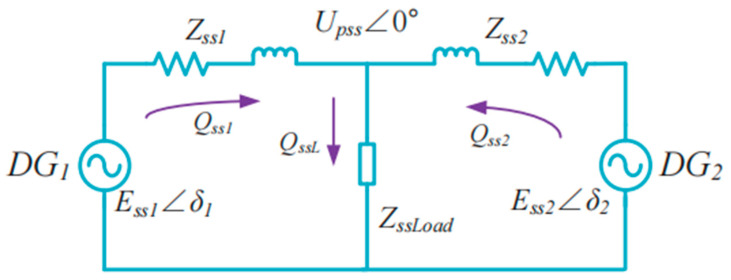
Equivalent circuit of two DG units in parallel with the frequencies of the injected SACSs.

**Figure 8 sensors-23-06269-f008:**
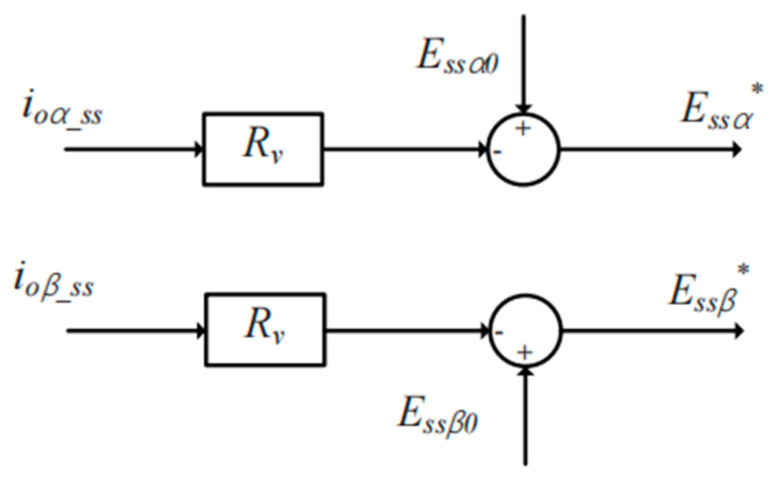
Block diagram of the virtual resistance control strategy for the injected SACS.

**Figure 9 sensors-23-06269-f009:**
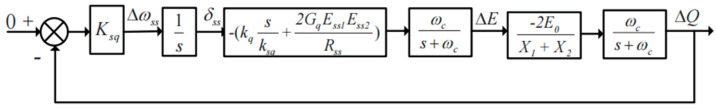
The control block diagram for reactive power sharing.

**Figure 10 sensors-23-06269-f010:**
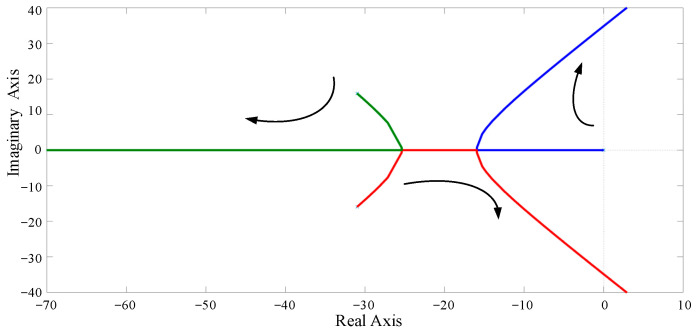
Root locus of the open-loop transfer function with the droop coefficient *k_sq_*.

**Figure 11 sensors-23-06269-f011:**
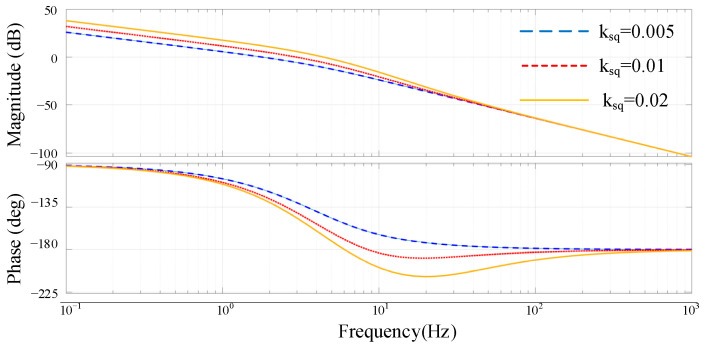
Bode diagram of the open-loop transfer function with droop coefficients *k_sq_* = 0.005, 0.01 and 0.02.

**Figure 12 sensors-23-06269-f012:**
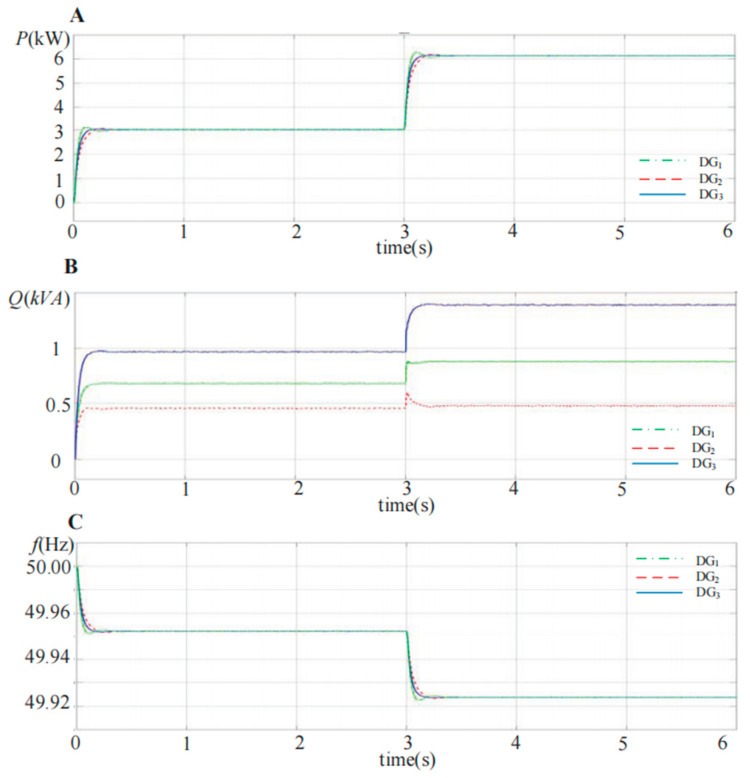
Simulation waveforms of the output active power *P*, reactive power *Q* and frequency f with the conventional droop control strategy for the three DG units in parallel. (**A**) Active power *P*; (**B**) reactive power *Q*; (**C**) fundamental frequency *f*.

**Figure 13 sensors-23-06269-f013:**
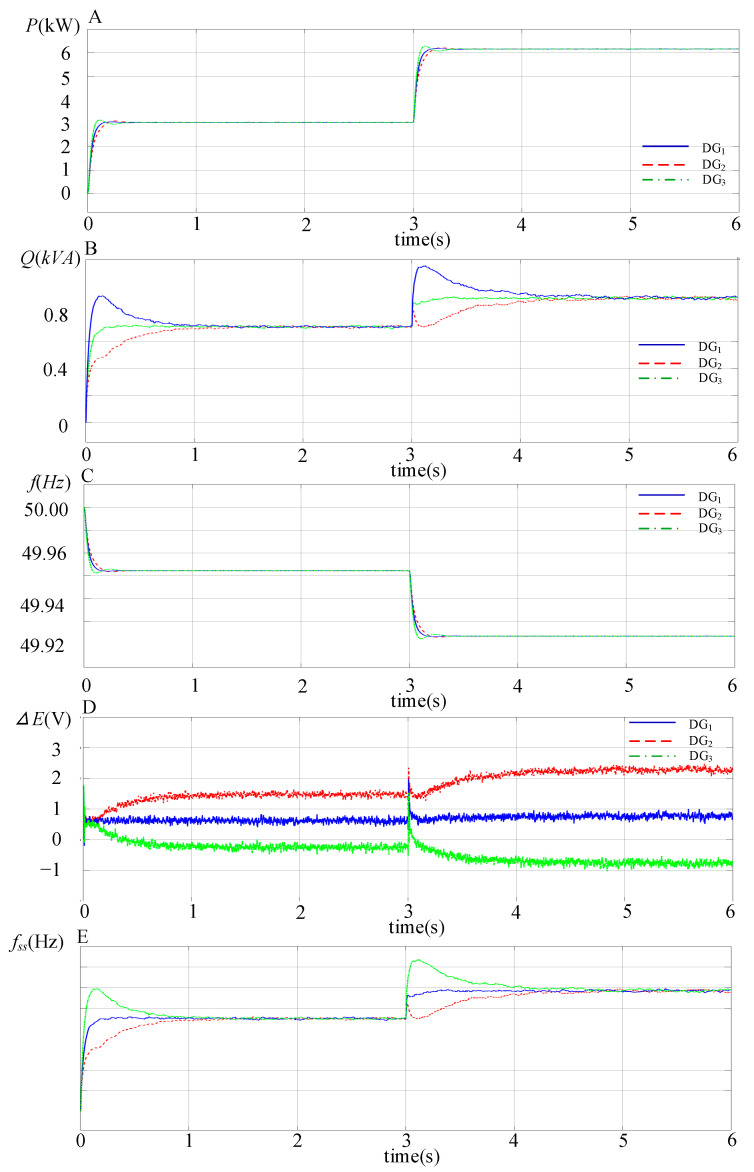
S Simulation waveforms of the output active power *P*, reactive power *Q*, frequency *f*, voltage compensator Δ*E* and SACS frequency *f_ss_* with the proposed modified droop control strategy for the three DG units in parallel. (**A**) Active power *P*; (**B**) reactive power *Q*; (**C**) fundamental frequency *f*; (**D**) voltage compensator AE; (**E**) SACS frequency *f_ss_*.

**Figure 14 sensors-23-06269-f014:**
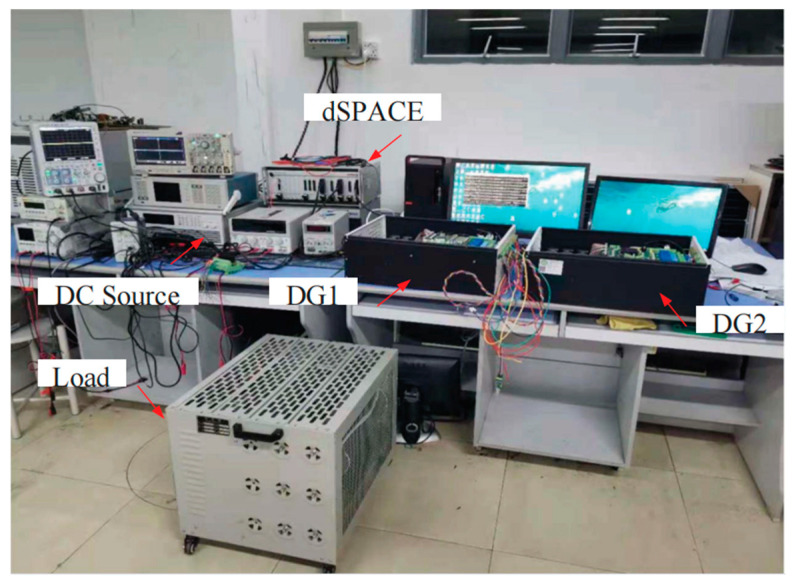
Experimental prototype for an islanded microgrid system.

**Figure 15 sensors-23-06269-f015:**
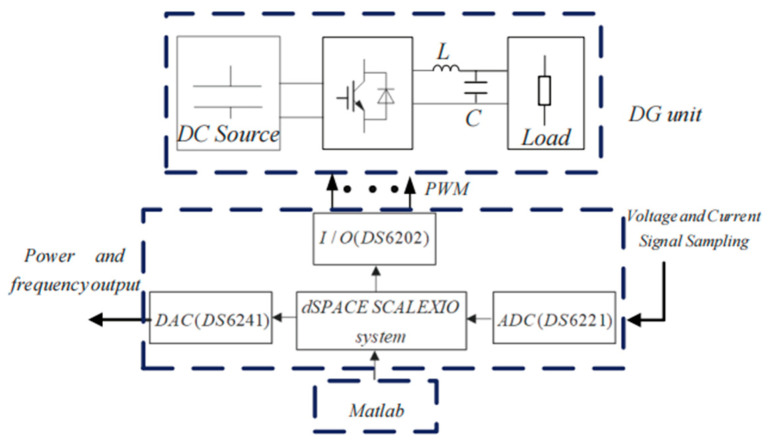
The dSPACE based experimental system.

**Figure 16 sensors-23-06269-f016:**
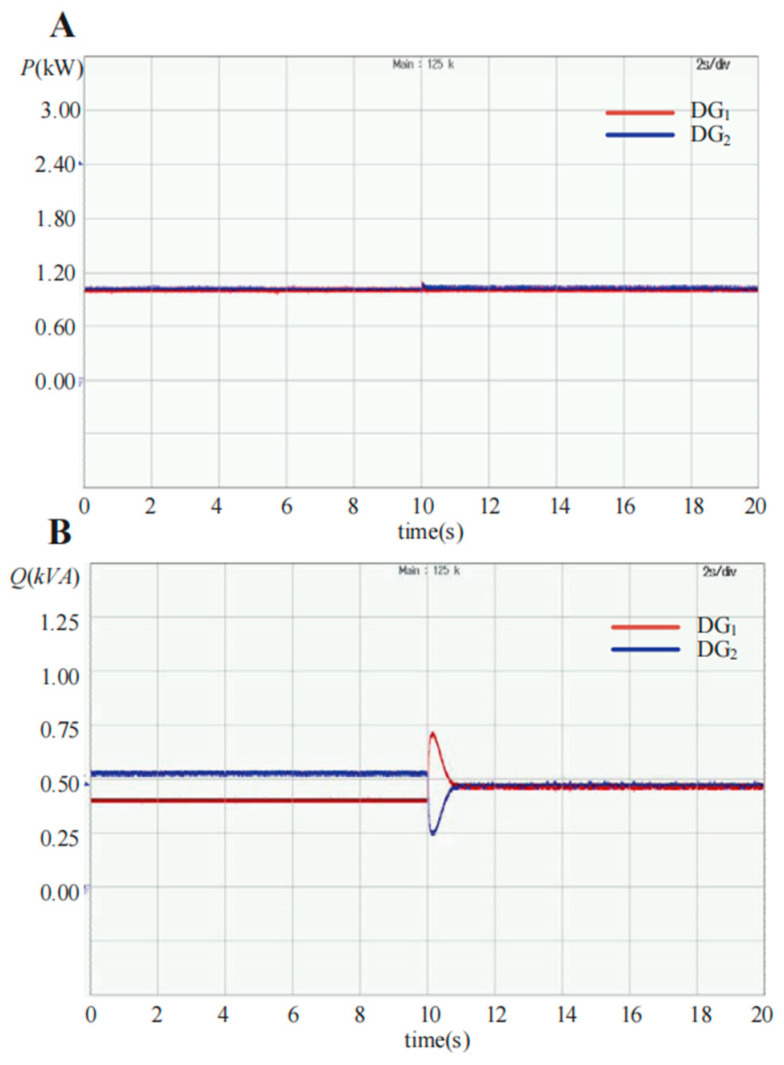
Experimental waveforms of the output active power *P* and reactive power *Q* changed from conventional droop control to the proposed modified droop control strategy with two DG units in parallel. (**A**) Active power *P*; (**B**) reactive power *Q*.

**Figure 17 sensors-23-06269-f017:**
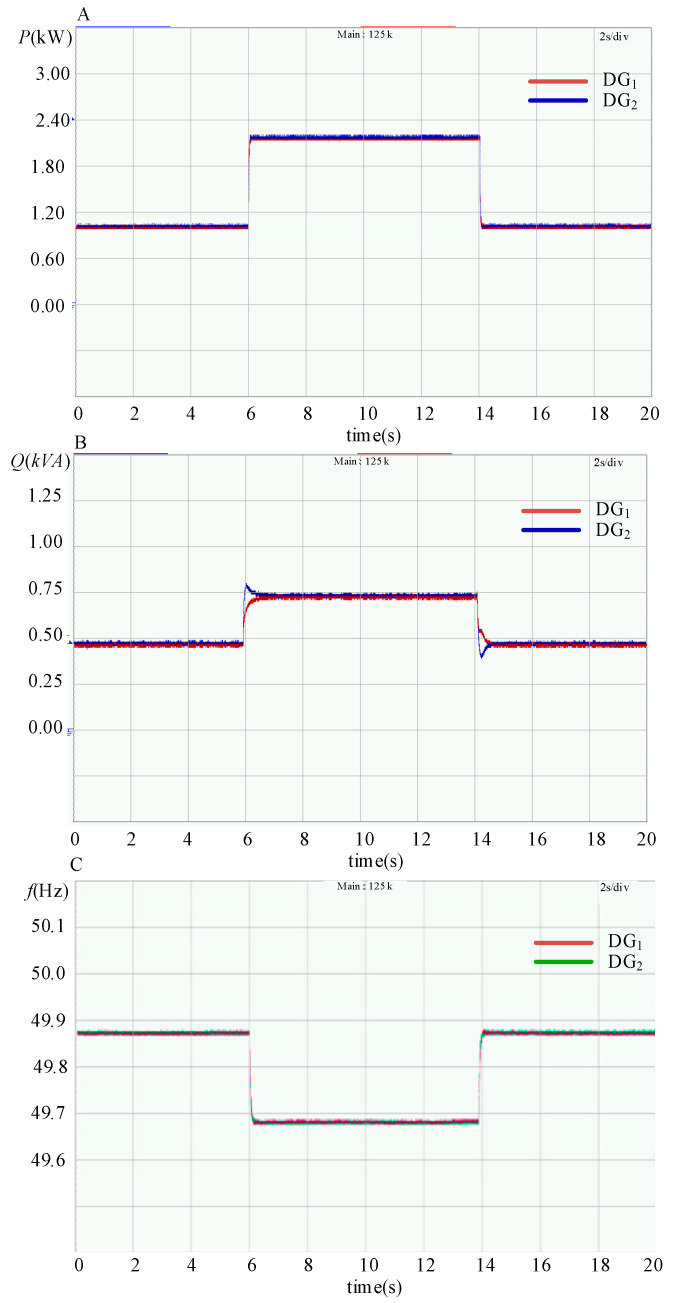
Experimental waveforms of the output active power *P*, reactive power *Q* and frequency *f* with the modified droop control strategy for the three DG units in parallel. (**A**) Active power *P*; (**B**) reactive power *Q*; (**C**) frequency *f*.

**Figure 18 sensors-23-06269-f018:**
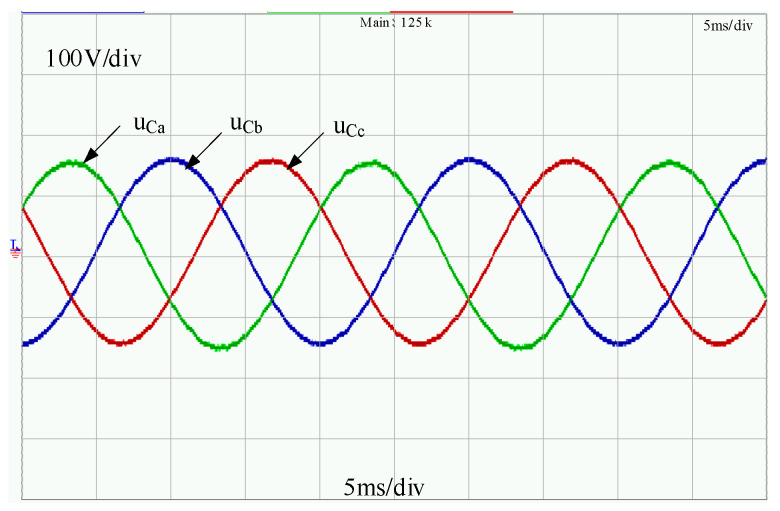
Experimental waveforms of the PCC voltage with the modified droop control strategy.

**Table 1 sensors-23-06269-t001:** Various control techniques, summaries and comparison.

Reference	Communication or Communication-Less Control Method	Computational Algorithm Burden	Accurate Active Power Sharing	Accurate Reactive Power Sharing
[12]	Communication	Medium	Yes	Yes
[13]	Communication	Medium	Yes	Yes
[14]	Communication	Medium	Yes	Yes
[15]	Communication	High	Yes	Yes
[16]	Communication	Low	Yes	Yes
[17]	Communication	High	Yes	Not considered
[18]	Communication	High	Yes	Yes
[19]	Communication-less	low	Yes	No
[20]	Communication-less	High	Yes	Yes
[21]	Communication-less	Medium	Yes	No
[22]	Communication-less	Medium	Yes	No
[23]	Communication-less	High	Yes	No
[24]	Communication-less	High	Yes	No
[25]	Communication-less	High	Yes	No
[26]	Communication-less	Medium	Yes	No
[27]	Communication-less	Medium	Yes	No
Proposed	Communication-less	Low	Yes	Yes

**Table 2 sensors-23-06269-t002:** System parameters.

Parameters	Value	
	Simulation	Experiment
Switching frequency *f_s_*	20 kHz	20 kHz
Fundamental frequency *f*	50 Hz	50 Hz
DC link voltage *U_dc_*	800 V	400 V
Fundamental voltage amplitude	311 V	155.5 V
Output filter inductance *L_f_*	1 mH	1 mH
Output filter capacitance *C_f_*	150 uF	150 uF
Feeder impedance of DG1(mH + Ω)	4 + 0.3	4 + 0.3
Feeder impedance of DG2(mH + Ω)	3.5 + 0.2	3.5 + 0.2
Feeder impedance of DG3(mH + Ω)	3 + 0.1	-
*P-ω* droop coeffificient *k_p_*	1.15 × 10^−4^	1 × 10^−3^
*Q-E* droop coeffificient *k_q_*	1 × 10^−3^	1.5 × 10^−3^
SACS voltage amplitude *E_ss_*_0_	2.5	1.25
SACS frequency *ω_ss_*	200 Hz	200 Hz
SACS droop coeffificient *k_sq_*	2 × 10^−3^	8 × 10^−3^
Gain *G_q_*	12	15
Low pass fifilters *ω_cp_*	31	31
Virtual resistor *R_v_* (Ω)	8	5
Load *R*_1_ (mH + Ω)	10 + 15	18 + 13.8
Load *R*_2_ (mH + Ω)	0 + 15	5 + 13.8

## Data Availability

The data that support the findings of this study are available from the corresponding author upon reasonable request.

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
