# Peer review of "Accurate Active and Reactive Power Sharing Based on a Modified Droop Control Method for Islanded Microgrids"

_sensors, 2023, doi:10.3390/s23146269_

Round 1

Reviewer 1 Report

Comments

The paper studied the Accurate Active and Reactive Power Sharing Based on a Modified Droop Control Method for Islanded Microgrids. It seems a hot topic, as many research papers have been published in this area. Below, I have provided my comments, which are as follows:

* It is not clear to me what the difference is between your study and the published study.

* What is the main contribution of your previous study to the current study? I need clarification on it.

* There are many grammatical issues in the paper.

* When you talk about published studies, you should provide references. See this sentence in your paper that you claimed, ". the error of reactive power sharing decreases,".

* The introduction and literature review needs to be comprehensively expanded, which confuses the author to follow you on what you plan to do with your paper.

* The literature review did not cover all studies close to your research area, which is a weakness of your study. Please summarize it in a table by adding more recent and relevant papers.

The abstract and conclusion must be revised by including rationale, solid reasoning, solutions, etc.

In this round of revision, I will go for a significant revision to see how the author responds to my comments.

Minor

Reviewer 2 Report

This paper proposes a modified droop control based on a small-ac-signal injection with the aim of achieving good P and Q power sharing among DGs in islanded microgrids. The authors claim that the main features of the proposal are: 1) that does not need communications; 2) any specific parameter of the feeder impedance is needed; 3) only local information is achieved.

The paper includes a “Parameter Design And Stability Analysis” section, simulations and experimental results.

Please review the following aspects:

1)           The authors compare their proposal with some references, especially with [15] and [16]. In line 61 they state: “Although the above mentioned methods proposed in [15] and [16] for each DG unit do not require the real-time information transmitted by communication line, however, the performance of power sharing is still largely dependent on these synchronous lines, which need to be connected between the DG units.”. What do you mean by “these synchronous lines”? This sentence is not clear and is complex for readers to understand. Considering that it is essential to understand the contribution of the work, I recommend rewriting it and clarifying the argument.

2)           Line 82 does not mention any problem of reference [22] that is overcome by the proposal presented in the manuscript.

3)           In line 89 the authors state that “1) most of the control strategies can only reduce the reactive power sharing error” as a negative aspect of references [17-23]. However, it is well known that active power sharing is not a big issue of the droop method since it can be fixed using a simple virtual impedance strategy. Please, complement this argument.

4)           Equation (12) is incorrect.

5)         The manuscript has many sentences that are not well written (I found more than 20). I recommend doing an in-depth review of the text. My final decision will depend on improving the writing of the next manuscript version.

The manuscript has many sentences that are not well written (I found more than 20). I recommend doing an in-depth review of the text. My final decision will depend on improving the writing of the next manuscript version.

Reviewer 3 Report

1- The abstract provides a clear summary of the paper, but it could be improved by mentioning the specific contribution of the proposed method in comparison to the existing literature on the topic

2-The introduction provides a good background on the topic, but it lacks a clear statement of the research gap and the specific research question that the proposed method aims to address. Also, some important papers like "https://doi.org/10.1016/j.seta.2021.101370" and   "https://doi.org/10.1016/j.seta.2021.101370" have not been studied. Please include them.

3- The methodology section provides a clear description of the proposed method, but it could be improved by providing a more detailed explanation of the technical aspects of the method, such as the small-ac-signal injection and the modified droop control strategy.

4-The simulation and experimental results section provide a thorough evaluation of the proposed method using both simulation and experimental data. However, it could be improved by providing more details on the performance metrics used in the evaluation and how they relate to the research question

5-The conclusion section provides a clear summary of the paper and the contributions of the proposed method, but it could be improved by discussing the limitations of the method and directions for future research.

Must be improved

Reviewer 4 Report

The paper entitled "Accurate Active and Reactive Power Sharing Based on a Modified Droop Control Method for Islanded Microgrids" is well written. Presented research results are up to the academic standards. The paper has a lot of merit and should be recommended for publication, but after correcting some important issues from the point of view of the quality of "Sensors" journal.

1. I would like to suggest authors to extension of the introduction so that this section can fully present the current state of art.

2. I also propose to add nomenclature (description of used symbols and abbreviations) to improve the readability of the article.

3. Although the topic and research results presented in the paper is current and interesting, the references section in the peer-revied paper is poor. The references section does not contain the current state of knowledge in the scope presented in the peer-revied paper.

4. Not all symbols in the equations are defined in the text. I suggest adding a nomenclature.

Round 2

Reviewer 3 Report

Thank you for addressing my comments 

Can be improved